Segmental analysis by speckle-tracking echocardiography of the left ventricle response to isoproterenol in male and female mice

Walsh-Wilkinson Elisabeth
Arsenault Marie
Couet Jacques jacques.couet@med.ulaval.ca
Universite Laval, Groupe de recherche sur les valvulopathies, Centre de recherche de l’Institut universitaire de cardiologie et de pneumologie de Quebec , Quebec , Canada
Gould Gwyn
Electronic publication date: 2021 Mar 12
Publication date: 2021
Volume: 9
Electronic Location ID: e11085
Received 2020 Nov 13; Accepted 2021 Feb 18
Copyright: ©2021 Walsh-Wilkinson et al.
Copyright year: 2021
Copyright holder: Walsh-Wilkinson et al.
License: This is an open access article distributed under the terms of the Creative Commons Attribution License, which permits unrestricted use, distribution, reproduction and adaptation in any medium and for any purpose provided that it is properly attributed. For attribution, the original author(s), title, publication source (PeerJ) and either DOI or URL of the article must be cited.
License URL: https://creativecommons.org/licenses/by/4.0/

Keywords: Speckle-tracking echocardiography, Isoproterenol, Takotsubo syndrome, Sex differences, Adrenergic suractivation, Strain, Mouse, Heart, Cardiac toxicity

Funding: Marie Arsenault from the Canadian Institutes of Health Research MOP-123186 IUCPQ Foundation This work was supported by an operating grant to Jacques Couet and Marie Arsenault from the Canadian Institutes of Health Research (MOP-123186) and the IUCPQ Foundation. The funders had no role in study design, data collection and analysis, decision to publish, or preparation of the manuscript.

==============================
We studied by conventional and speckle-tracking echocardiography, the response of the left ventricle (LV) to a three-week continuous infusion of isoproterenol (Iso), a non-specific beta-adrenergic receptor agonist in male and female C57Bl6/J mice. Before and after Iso (30 mg/kg/day), we characterized LV morphology and function as well as global and segmental strain. We observed that Iso reduced LV ejection in both male (−8.7%) and female (−14.7%) mice. Several diastolic function parameters were negatively regulated in males and females such as E/A, E/E′, isovolumetric relaxation time. Global longitudinal (GLS) and circumferential (GCS) strains were reduced by Iso in both sexes, GLS by 31% and GCS by about 20%. For the segmental LV analysis, we measured strain, strain rate, reverse strain rate, peak speckle displacement and peak speckle velocity in the parasternal long axis. We observed that radial strain of the LV posterior segments were more severely modulated by Iso than those of the anterior wall in males. In females, on the other hand, both posterior and anterior wall segments were negatively impacted by Iso. Longitudinal strain showed similar results to the radial strain for both sexes. Strain rate, on the other hand, was only moderately changed by Iso. Reverse strain rate measurements (an index of diastolic function) showed that posterior LV segments were negatively regulated by Iso. We then studied the animals 5 and 17 weeks after Iso treatment. Compared to control mice, LV dilation was still present in males. Ejection fraction was decreased in mice of both sex compared to control animals. Diastolic function parameters, on the other hand, were back to normal. Taken together, our study indicates that segmental strain analysis can identify LV regions that are more negatively affected by a cardiotoxic agent such as Iso. In addition, cessation of Iso was not accompanied with a complete restoration of cardiac function after four months.

Introduction

Adrenergic overstimulation using isoproterenol (Iso) is often used to induce cardiac toxicity in small rodent models (Gomes et al., 2013; Chang et al., 2018; Kudej et al., 1997). Sometimes described as an HF model (systolic and/or diastolic), a cardiac hypertrophy model or a SIC model (Takotsubo-like syndrome)  (Puhl et al., 2016; Wallner et al., 2016; Sachdeva, Dai & Kloner, 2014; Shao et al., 2013a; Shao et al., 2013b; Angelini & Gamero, 2019), Iso treatment of rats or mice has been the object of an important amount of literature.

Many regimens of Iso administration to rodents have been used in past studies such as a single high dose bolus, several injections over days as well as continuous infusion using osmotic micro-pumps implanted subcutaneously or intra-peritoneally, or micro-pellets at different dosages and for various duration (Chang et al., 2018; Shao et al., 2013b; An et al., 2016; Ali et al., 2019; Ma et al., 2011). This diversity of experimental set-ups makes the comparison between studies sometimes difficult, as the goals pursued by the authors were often different. In addition, as for many preclinical studies, most experiments were conducted in male animals and less often in females. In spontaneously hypertensive rats, it was observed that Iso treatment was more prone in induce LV dilation in males than in females (Michel et al., 2017). In mice, more studies were performed using females but again, few studies studied in parallel both sexes (Zhu et al., 2016).

Sex-related differences in the Iso model have been studied using echocardiography in C57Bl6 mice infused for 7 or 14 days (10 mg/kg/day) but relatively few differences in either systolic or diastolic cardiac function were found. Most of the differences were identified at tissue level (Zhu et al., 2016). More recently, another study found no sex differences in mice treated with Iso for 14 days (Grant et al., 2020).

In the present study, we wished to document the left ventricular (LV) response to a beta-adrenergic receptor-mediated insult using the Iso mouse model and to identify possibly morphological and functional sex differences. We used a relatively high dosage of Iso (30 mg/kg/day) for a longer period (21 days). Using conventional and speckle tracking echocardiography (STE), we evaluated if LV dysfunction was present and then performed a regional study of this dysfunction. We also studied how male and female mice echo parameters evolved 5 and 17 weeks post-Iso.

Our results indicate that Iso induced both systolic and diastolic function impairments in mice and that only small sex differences were present in the extent of these. By STE, we showed that the response to Iso is not homogeneously distributed throughout the LV walls as the posterior LV wall seems more sensitive to Iso effects, at least in male mice. As for LV reverse remodeling taking place after Iso, dilation in males was still present four months later.

Materials and Methods

Mouse model

The present study was conducted within the Mouse Animal model of Sex Differences and Aging in heart Failure (MASDAF) study, which follows longitudinally C57Bl6/J mice up to two years in order to investigate biological sex and aging effects on the LV response to an insult. In this sub-study, we compared the LV response of both males and females (age 8 weeks) from Jackson Laboratory (Bar Harbor, ME, USA). After a week of acclimatization, micro-osmotic pumps (cat. no.: 1004; Alzet, Cupertino, CA, USA), releasing isoproterenol (Iso: 30 mg/kg/day; Sigma-Aldrich, Mississauga, Ont, Canada; n = 10 mice/group) or vehicle (saline; Ctrl. n = 8 mice/group) were implanted subcutaneously in the back of the neck and left for 21 days (Roussel et al., 2018). Experienced technicians for health and behavior monitored the animals daily during the protocol. The animals were weighed weekly. No mouse displayed markers associated with death or poor prognosis of quality of life, or specific signs of severe suffering or distress, which would have led to early and immediate euthanasia. Among those, significant loss or gain of weight, grooming and changes in behavior were monitored. The protocol was approved by the Université Laval’s animal protection committee and followed the recommendations of the Canadian Council on Laboratory Animal Care (protocol #2019-360, VRR-19-075).

Echocardiography

An echocardiography (Echo) exam was performed the day before Iso infusion started and three weeks later. Echo images were acquired on a Vevo 3100 imaging system (VisualSonics, FujiFilm, Toronto, Canada) by the same investigator and analyzed off-line using Vevo LAB software. The investigator was blinded for animal identification. Animals were anesthetized and placed in a supine position on a heated platform (39 °C). The concentration of isoflurane was maintained around 1.5–2.5%, so that the heart rate was kept over 400 beats/minute.

2D echo.

M-mode images were recorded to measure diastolic LV walls thickness and chamber diameters from the parasternal long axis (PSLAX) view and the short axis (SAX) view at the papillary muscle level. LV mass was calculated by the VevoLab echo analysis software (VisualSonics) using the following equation: 1.053 × [(EDD + PW + IVSW)3 − EDD3] × 0.8. EDD: end-diastolic LV diameter, PW: posterior diastolic wall thickness and IVSW: interventricular septal diastolic wall thickness. Fractional shortening from M-mode images was calculated using the following equation: (EDD − ESD)/EDD where: ESD end-systolic LV diameter. Mitral flow was measured by pulsed wave Doppler from an apical four-chamber view. Early diastolic peak filling velocity (E wave), peak filling velocity at atrial contraction (A wave), E wave deceleration time and the E/A ratio were obtained. The early-diastolic peak velocity (E′), the late-diastolic peak velocity (A′) of the mitral valve annulus and E′∕A′ as well as E/ E′ were obtained using tissue Doppler. LV volumes, ejection fraction (EF), stroke volume and cardiac output were calculated using the Simpson’s rule method from LV chamber area tracings.

Speckle-tracking echo.

2D echo B-mode loops were acquired from the LV PSLAX and analyzed using Vevo Strain software (VisualSonics). Images were acquired at the highest frame rate possible (232 frames/s) and strain analysis was performed in the radial and longitudinal axes as previously described  (Walsh-Wilkinson et al., 2020). Also calculated by the Vevo Strain software were the reverse strain rate (a diastolic function index), speckle displacement (mm) and velocity (mm/s). Figure 1A illustrates the various segments studied as well as the direction of strain or speckle displacement.

Figure 1 Speckle tracking strain analysis.

Radial and longitudinal strains can be obtained using the parasternal long axis (PSLAX) view. The six segments are also identified. Characters colors correspond to those used in the graphs for each of these LV segments. Ant, anterior, Post, posterior. Before after effects of Iso on the strain. Radial (A–B) and longitudinal (C–D) peak strains were obtained using the parasternal long axis view. The six segments are identified. Characters colors correspond to those used in the graphs for each of these LV segments.Radial (E–F and G–H) and longitudinal (I–J and K–L) peak strains were grouped either as from the base, the midsection or the apex or as from the anterior or the posterior segments in males (left panels) and females (right panels). Ant, anterior, Post, posterior. Males are represented on the left panels and females on the right. Results are represented as violin plots ( n = 8–10). Inner black lines represent quartiles of the data. Significance between groups was calculated with paired Student’s T-test. *, p < 0.05, **, p < 0.01, ***, p < 0.001 and ****, p < 0.0001 between corresponding pre-Iso and Iso animals.

Statistical analysis

All data are expressed as mean ± standard error of the mean (SEM). Statistical analyses were performed on the log10 of the data. Normality was assessed using the Shapiro–Wilk test. Intergroup comparisons were conducted using Student’s T-test using GraphPad Prism 8.4, (GraphPad Software Inc., La Jolla, CA, USA). Data from Tables 1–3 were analyzed using 2-way ANOVA and Holm-Sidak pos t-test. P < 0.05 was considered statistically significant.

Table 1 Left ventricle morphology and systolic function in male and female mice receiving isoproterenol or not for three weeks.

Parameters	M Ctrl, N = 8	M Iso, N = 10	F Ctrl, N = 8	F Iso, N = 10	Sex	Iso	Sex × Iso	
BW, g	25.0 ± 0.31	27.9 ± 0.53b	19.9 ± 0.28	20.8 ± 0.45	<0.0001	ns	ns	
Tibia, mm	21.0 ± 0.08	21.7 ± 0.10d	20.5 ± 0.08	20.3 ± 0.13	<0.0001	ns	ns	
M-mode		
EDD, mm	3.8 ± 0.03	3.9 ± 0.07	3.6 ± 0.06	3.6 ± 0.07	<0.0001	ns	ns	
ESD, mm	2.6 ± 0.08	2.9 ± 0.11a	2.3 ± 0.05	2.6 ± 0.07b	<0.01	<0.001	ns	
IVS, mm	0.82 ± 0.02	0.82 ± 0.02	0.76 ± 0.01	0.77 ± 0.02	<0.001	ns	ns	
PW, mm	0.84 ± 0.01	0.83 ± 0.02	0.78 ± 0.01	0.75 ± 0.02	<0.0001	ns	ns	
RWT	0.44 ± 0.01	0.42 ± 0.01	0.43 ± 0.01	0.42 ± 0.01	ns	ns	ns	
FS, %	32.2 ± 1.69	24.8 ± 1.52b	35.4 ± 0.98	27.5 ± 0.86c	<0.05	<0.0001	ns	
LV, mg	86 ± 2.1	95 ± 4.1	76 ± 2.6	75 ± 2.7	<0.0001	ns	ns	
iLV, mg/g	3.4 ± 0.10	3.4 ± 0.11	3.8 ± 0.14	3.6 ± 0.11	<0.01	ns	ns	
Simpson’s		
EDV, µl	54.6 ± 1.83	66.9 ± 3.62b	44.6 ± 0.74	51.0 ± 2.45	<0.0001	<0.001	ns	
ESV, µl	23.4 ± 0.72	34.8 ± 2.78b	17.1 ± 0.34	27.4 ± 2.13b	<0.01	<0.0001	ns	
SV, mm	31.2 ± 1.39	32.1 ± 1.91	27.5 ± 0.56	23.6 ± 1.19	<0.001	ns	ns	
EF, %	57.1 ± 1.01	48.4 ± 2.19a	61.7 ± 0.55	47.0 ± 2.80c	ns	<0.0001	ns	
HR, bpm	494 ± 17.9	415 ± 10.0b	441 ± 16.8	448 ± 9.0	ns	<0.05	<0.01	
CO, ml/min	15.3 ± 0.69	13.3 ± 0.73	12.2 ± 0.56	10.6 ± 0.60	<0.0001	<0.01	ns	
Notes.

Ctrl control

Iso isoproterenol treatment

BW body weight

LV left ventricle mass

EDD end diastolic LV diameter

ESD end-systolic diameter

IVS interventricular septum

PW posterior wall

RWT relative wall thickness

FS fractional shortening

EDV end-diastolic volume

ESV end-systolic volume

SV stroke volume

EF ejection fraction

HR heart rate

bpm beats per minute and CO: cardiac output

Values are expressed as the mean ± SEM. Two-way ANOVA statistical analysis results are displayed for each factor, sex and Iso, respectively. Intergroup p values were calculated using Holm-Sidak post-test.

a p < 0.05.

b p < 0.01.

c p < 0.001.

d p < 0.0001 between Ctrl and Iso groups, respectively.

ns, not significant.

Results

Isoproterenol induces systolic and diastolic impairments in male and female mice

Before implantation of the micro-osmotic pump, a complete echocardiography exam was performed for each animal. Baseline values are listed in Table S1. With the exception of those related to the relative size of the animals depending on their biological sex, systolic and diastolic function baseline parameters were mostly similar between males and females.

Ten of these eighteen mice of each sex were then treated for three weeks with a continuous infusion of isoproterenol (Iso), a non-specific beta-adrenergic receptor agonist. On day 21, osmotic pumps were removed and the day after, a second echo exam was performed and data were compared to controls (Ctrl). As depicted in Table 1, Iso treatment had relatively similar effects in male and female mice on M-mode measurements. End systolic LV diameter (ESD) was increased in mice of both sexes resulting in a corresponding decrease in fractional shortening (−7.4% for males and −7.9% for females). SAX M-mode data were similar (not shown but included in the raw data). Both end-diastolic (EDV) and end-systolic volumes (ESV) were increased after Iso treatment, respectively by 23% and 49% in males and by 14% (not significant; ns) and 60% in females. This resulted in lowered ejection fraction for both male (−8.7%) and female (−14.7%) mice compared to control. Stroke volume was unchanged in males but tended to decrease in females (ns). Cardiac output was lower in Iso-treated mice.

Iso treatment also influenced diastolic echo parameters (Table 2). The E wave measured by pulse-wave Doppler was decreased in male mice but not in females. We then measured E′ and A′ waves from the mitral valve annulus movements by tissue Doppler. The E′ wave was significantly reduced by 34% in males, whereas the E/E′ ratio was significantly increased by Iso (30%). Only a tendency was registered for females (+20%). Isovolumetric relaxation time (IVRT) was significantly longer in both Iso groups compared to controls (+40% in males and +20% in females, respectively).

Table 2 Left ventricle diastolic parameters in male and female mice receiving isoproterenol or not for three weeks.

Parameters	M Ctrl, N = 8	M Iso, N = 10	F Ctrl, N = 8	F Iso, N = 10	Sex	Iso	Sex × Iso	
E, mm/s	662 ± 23.6	551 ± 18,4c	578 ± 15.6	579 ± 14.2	ns	<0.01	<0.01	
A, mm/s	430 ± 21.7	380 ± 9.0	363 ± 10.1	375 ± 16.9	<0.05	ns	ns	
E/A	1.55 ± 0.03	1.45 ± 0.03	1.60 ± 0.05	1.57 ± 0.07	ns	ns	ns	
E dec, ms	20.0 ± 1.20	25.2 ± 1.65	18.8 ± 0.70	22.4 ± 1.88	ns	<0.01	ns	
E′, mm/s	27.6 ± 1.49	18.3 ± 1.04d	26.1 ± 1.61	22.3 ± 0.81	ns	<0.0001	ns	
A′, mm/s	19.3 ± 0.44	13.8 ± 0.98c	20.4 ± 1.41	16.4 ± 0.49a	ns	<0.0001	ns	
E∕E′	24.2 ± 1.45	31.4 ± 1.57a	22.3 ± 1.42	30.5 ± 2.54	<0.05	<0.001	ns	
E′/A′	1.43 ± 0.06	1.36 ± 0.04	1.29 ± 0.04	1.43 ± 0.06	ns	ns	ns	
IVRT, ms	15.1 ± 0.32	21.1 ± 0.57d	16.1 ± 0.61	19.4 ± 0.93b	ns	<0.0001	<0.05	
Notes.

Dec deceleration time

IVRT Isovolumetric relaxation time

Values are expressed as the mean ± SEM. Two-way ANOVA statistical analysis results are displayed for each factor, sex and Iso, respectively. Intergroup p values were calculated using Holm-Sidak post-test.

a p < 0.05.

b p < 0.01.

c p < 0.001.

d p < 0.0001 between Ctrl and Iso groups, respectively.

ns, not significant.

In Table 3 is illustrated the evolution of global longitudinal (GLS) and global circumferential (GCS) strains in male and female mice receiving Iso. Global strain measurements take into consideration the entire LV wall comparing the relative changes in LV inner contour length during the cardiac cycle. GLS is calculated from the PSLAX view and GCS from SAX. For both parameters, GLS and GCS, more negative values are associated with better fractional change. In control animals, GLS and GCS values were similar between the sexes. After 3 weeks of Iso infusion, GLS values became significantly less negative in both males (+31%) and females (+31%). As for GCS, it worsened similarly in males (+17%) and females (+23%) after Iso.

Table 3 Global LV strain parameters in male and female mice treated with isoproterenol or not for three weeks.

Parameters	M Ctrl, N = 8	M Iso, N = 10	F Ctrl, N = 8	F Iso, N = 10	Sex	Iso	Sex × Iso	
GLS	−19.1 ± 0.6	−13.1 ± 0.8d	−20.3 ± 0.8	−14.0 ± 0.9c	ns	<0.0001	ns	
GCS	−27.6 ± 1.5	−18.3 ± 1.0d	−26.1 ± 1.6	−22.2 ± 0.8	ns	<0.0001	<0.05	
Notes.

GLS global longitudinal strain

GCS global circumferential strain

Values are expressed as the mean ± SEM.

Two-way ANOVA statistical analysis results are displayed for each factor, sex and Iso, respectively.

Intergroup p values were calculated using Holm-Sidak post-test.

c p < 0.001.

d p < 0.0001 between Ctrl and Iso groups, respectively.

ns, not significant.

Segmental analysis using speckle tracking echocardiography (STE) points toward a non-uniform LV response to Iso in males

The LV was divided into six segments for the PSLAX view, as described in Fig. 1A. This allowed us to investigate if the effects of Iso infusion were distributed globally along the LV endocardial wall or if one or many segments were more seriously affected than others were.

Using segmental analysis, we compared the strain at baseline (Pre-Iso) and after Iso in male and female mice. We chose to use baseline values as control in order to limit the effects of intra-group variability between animals. As illustrated in Figs. 1B–1E, radial and longitudinal strains were reduced by Iso in males for all three posterior LV wall segments, whereas anterior segments strain was mostly unchanged. Radial strains of two posterior wall segments (base and mid) was also reduced in females. In addition, radial strain for all LV anterior wall segments was reduced. Average radial and longitudinal strains (all segments; in black) were negatively modulated by Iso for both male and female mice.

We grouped data from the segmental strain analysis as from either the base, the mid-ventricle or the apex regardless of the posterior or the anterior walls. We did the same for the anterior or the posterior wall data that were analyzed together regardless if they originated from the base, mid-ventricle or apex (Kudej et al., 1997). As illustrated in Figs. 1E–1M, both radial and longitudinal strains were reduced for the base LV segment in male and female mice. Mid LV radial strain was reduced for both sexes. This was also the case for the longitudinal strain in males as well as the apex longitudinal strain in male and female animals. We then compared if the posterior wall strain was more affected by Iso than the anterior wall. In females, Iso decreased both radial and longitudinal strains to the same extent (−30%). In males, the radial strain was significantly decreased only for the posterior wall. The longitudinal strain was reduced for both walls but this decrease was more important for the posterior wall than the anterior one (−39% vs. −18%, respectively).

We measured the indexed time-to-peak (T2P) for strain data. This represents the time for strain to reach its maximal value from baseline (R-wave). We indexed the value for the duration of the cardiac cycle (1/HR) in order to take into account variations in heart rate between animals. Values are thus expressed as a percentage of a cardiac cycle for strain to reach its peak. As illustrated in Fig. S1, iT2P for each LV segment in radial and longitudinal remained mostly stable in males. In females however, a greater variation after Iso was observed suggesting that normal LV wall synchronicity was perturbed. We then calculated the standard deviation (SD) of iT2P for each 6 segments. In males (blue), these parameters remained stable after Iso but in females (orange), radial iT2P SD was increased.

As illustrated in Video S1 (males) and Video S2 (females), LV wall deformation after Iso was markedly reduced. When LV movement tendency was expressed using velocity vectors as illustrated in Fig. 2, it can be appreciated that in both systole and diastole, the reduction of the length and the changes in the orientation of velocity vectors induced by Iso.

Figure 2 PSLAX LV wall trace tendency before and after Iso.

LV wall trace tendency is expressed as velocity vectors for >48 points around the endocardium in systole (left panels) and diastole (right panels) before and after Iso in males (top panels) and female mice (bottom panels). Images of velocity vectors corresponds to the maximal peak (systole) and minimal peak (diastole) of the average curve of all six segments curves for speckle velocity in PSLAX. As evidenced by these PSLAX B-mode views, velocity vector orientation and length varies during the cardiac cycle. Vertical (radial) and horizontal (longitudinal) components of each vector do not correspond necessarily to the respective peak value of each orthogonal component.

Peak strain rate (SR) represents a systolic function index that indicates the maximal rate of deformation (strain) during systole. On the other hand, reverse peak SR happens during the passive LV filling phase of diastole (Fig. 3A) and has been proposed as a new index of diastolic function (Schnelle et al., 2018). Changes in radial and longitudinal strain rates (SR) caused by Iso were relatively minor and concentrated on the posterior segments in males and females (Figs. 3B–3E). Peak reverse SR showed a pattern reminiscent of those of strain and SR. Iso negatively modified LV posterior segments in both males and females (Figs. 3F–3H). Several anterior wall segments were also affected, mostly in females.

Figure 3 Peak SR and Peak reverse SR (rSR).

In the background is represented a screen caption of an M-mode loop of three cardiac cycles. In green, the EKG is superposed at the top. Radial strain rate curves are depicted for each SAX segment and an “average” curve in black. Notice that all curves begin at the R wave of the EKG. As evidenced by this representation, the first SR peak (top yellow circle) corresponds to the maximal SR (1/s) whereas the second peak (bottom yellow circle) is inverted and happens during the early stage of LV relaxation as evidenced by the M-mode image underneath. A male mouse after three weeks of Iso is represented. B-H. Before after effects of Iso on strain rate (SR) and reverse strain rate (rSR). Radial (B–C) and longitudinal (D–E) peak SR and rSR (F–H) are illustrated. Characters colors correspond to those used in the graphs for each of these LV segments. Ant, anterior; Post, posterior; SW, septal wall and FW, free wall. Males are represented on the left panels and females on the right. Results are represented as violin plots (n = 8–10). Inner black lines represent quartiles of the data. Significance between groups was calculated with paired Student’s T-test. *, p < 0.05, **, p < 0.01 and ***, p < 0.001 between corresponding pre-Iso and Iso groups.

Peak speckle displacement and peak speckle velocity are analogous to strain and strain rate although they are not expressed relative to a second point in the myocardium as for the strain and SR. Original speckle position or speckle velocity is determined at the R-wave and is arbitrarily fixed to zero. Iso reduced radial speckle displacement and velocity mainly for the posterior wall segments in males and females compared to baseline values. Displacement was also decreased for the anterior wall in females. In the longitudinal direction, peak speckle displacement remained stable in males and was reduced for only one segment (anterior base) in females. A similar situation was observed for longitudinal velocity of the LV wall (Fig. S2).

After 21 days (week 3) of Iso treatment, the osmotic micro-pumps were removed and the LV reverse remodeling was studied by echo at week 8 and 20 or 5 and 17 weeks post-Iso, respectively. As illustrated in Figs. 4A–4J, LV end-systolic volume in males remained increased compared to control animals, whereas in females, no difference was present. Iso cessation helped reduce LV end-systolic volume in both males and females. However, only females returned to normal values at week 8 before increasing again 12 weeks later. This resulted that both male and female mice had a decreased ejection fraction four months post-Iso. On the other hand, both diastolic parameter, E/E′ and IVRT, were normalized 5 weeks after Iso treatment. At the end of the follow-up, GLS and GCS had returned to normal for males but only GCS for females (Figs. 4K–4L). Segmental strain analysis showed a similar picture for males where strain values were unchanged compared to age-matched controls. On the other hand, both radial and longitudinal strains showed abnormalities in females 17 weeks after Iso treatment (Fig. 4M). Posterior apical segment in radial and both apical segments showed diminished strain (Figs. 4O–4R). We also look at the reverse strain rate as a diastolic parameter. Compared to control animals, average reverse SR in Iso-treated animals at week 20 were normal (Fig. 4N).

Figure 4 Reversibility of Iso treatment.

(A–J) Head-to-head longitudinal comparison of several echo parameters of control (open circles) and Iso-treated (closed circles) male (blue) and female (orange) mice. EDV, end-diastolic volume; ESV, end-systolic volume; EF, ejection fraction and IVRT, Isovolumetric relaxation time. *, p < 0.05, **, p < 0.01, ***, p < 0.001 and ****, p < 0.0001 between corresponding control and Iso animals using unpaired Student’s T-test. #, p < 0.05; ##, p < 0.01; ###, p < 0.001 and ####, p < 0.0001 between measurements at different times using ANOVA and Tukey’s pos t-test. K–L, Global longitudinal strain (GLS) and global circumferential strain (GCS) at week 20 in male and female mice treated or not with Iso for 3 weeks. M–N, Average radial and longitudinal peak strains at week 20. O–R, Radial and longitudinal peak strains at week 20 in male and female mice treated or not with Iso. The six segments are identified. Results are represented as violin plots (n = 8–10). Inner black lines represent quartiles of the data. Significance between groups was calculated with unpaired Student’s T-test. *, p < 0.05; **, p < 0.01 and ***, p < 0.001 between corresponding control (Ctrl) and Iso animals.

Discussion

In the present study, we demonstrated that the LV response to beta-adrenergic sur-activation using Iso, resulted in both systolic and diastolic function impairments in mice of both sexes. Using speckle-tracking echocardiography, we proceeded to a thorough investigation of the strain, strain rate and other related parameters. In males, we observed that the LV posterior wall was in general, more negatively affected by Iso than the anterior wall. This was true for the strain in both radial and longitudinal directions. In females, strain in the LV posterior wall was also negatively reduced but the anterior wall was also affected making the effects of Iso more global.

Most baseline values for the different STE parameters (strain, SR, and rSR) were similar between males and females. Radial peak speckle displacement and velocity for several segments were significantly smaller in females, which is likely related to a smaller heart size. To our knowledge, this study is the first to report segmental LV wall displacement and velocity data from normal young adult mice.

Conventional echocardiography

Systolic function as estimated by the ejection fraction (EF) was relatively more reduced in females than in males. Baseline EF values were higher in females, though. Various methods are available for the evaluation of EF by echo. The availability for small animals of four-dimensional (4D) echo LV chamber reconstruction over the cardiac cycle adds another way to estimate EF. In this study, we finally opted for the Simpson’s method from LV chamber area tracings in PSLAX instead of 4D echo. Four-dimensional echo, in our hands, probably underestimated LV volume, which lead to incorrect calculations of LV volumes, stroke volume, ejection fraction and so forth. Depending on the method, EF estimates went from being in the 65–70% range using M-mode, 57–62% range using Simpson’s method to 55–60% (PSLAX) and 45–50% (SAX) using 4D echo in normal mice. In addition, Iso effects seemed to be masked using 4D echo and variability was high using this method.

Several factors can limit acquisition of quality 4D echo scans in rodents as described before in several studies (Grant et al., 2020; Grune et al., 2018; Damen et al., 2017; Rutledge et al., 2020). One is interference from anatomical structures (sternum, ribs and lungs) that often obscure parts of the heart, making it difficult to visualize and to trace LV walls. Working with those low-quality 4D echo scans can significantly increase intra-observer variability and thus, reduce reproducibility.

Despite these discrepancies between the methods used to evaluate EF, Iso increased both systolic LV diameters and volumes in mice. A LV dilation was also apparent in 2D B-mode views, whereas the wall thickness seemed to remain remained stable (see Fig. 2, Videos S1 and S2). This LV remodeling helped preserve cardiac output although a trend for a decrease was present. It is difficult to conclude that mice were experiencing symptoms of HF, especially since cardiac output was mostly preserved. HF in small rodents is often recognized by increased lungs weight after euthanasia or decreased exercise performance (Gomes et al., 2013). We did not test resistance to forced exercise in our animals.

In the case of diastolic parameters measured by either pulse-wave or tissue Doppler, many of them were negatively modulated by Iso in male mice. In females, only those measured by tissue Doppler were affected by Iso and, to a lesser extent than for males. As in humans, defining diastolic dysfunction by echo is difficult although clearly, Iso treatment caused diastolic function impairments, especially in male mice.

Increased myocardial interstitial fibrosis is often related to diastolic dysfunction. Iso treatment has been shown to induce intertitial fibrosis in mice in other studies. Collagen production was shown to by more important in male mice receiving Iso than in females  (Zhu et al., 2016). Among the other mechanisms that have been proposed to explain HF induced by Iso is an increased rate of cardiomyocyte apoptosis (Zhuo et al., 2013). Sur-activation of the beta-adrenergic system by Iso imposes an increased cardiac workload, which is associated with increased consumption of oxygen by the myocardium. This can lead to increased production of reactive oxygen species (Ma et al., 2015). An additional mechanism for negative effects of Iso is lipotoxicity. In an acute model of Iso-induced stress cardiomyopathy, rapid lipid accumulation was noticed as soon as two hours after injection. It is not clear if here, in our chronic setting, that this intracellular lipid accumulation is present or lasts for days but this does not exclude a possible toxicity of Iso for the cardiac myocytes leading to apoptosis (Shao et al., 2013b).

Speckle-tracking echocardiography

In this study, conventional echo highlighted LV anomalies both morphological and functional. Our study design did not aim at testing if strain analysis was more sensitive for early detection of dysfunction as it was done previously (An et al., 2016; Li et al., 2014; Peng et al., 2009; Szymczyk et al., 2013). Our goal was to evaluate if additional and valuable information could be obtained using STE such as segmental strain and SR but also reverse SR, speckle displacement and speckle velocity.

Using the strain analysis, we observed that the LV posterior wall was in general, more negatively modulated by Iso than the anterior wall. The base and the mid-ventricle segments were the most affected and the apex, to a lesser extent. This trend was clearer in males although it was also present in females. Obviously, Iso treatment being longer and more severe in this study, triggered a compensatory response (An et al., 2016). LV dilatation had time to take place in males, which had to be accompanied by a concomitant extracellular matrix remodeling. Fibrosis has been described in other studies looking at Iso effects on the mouse myocardium (Grant et al., 2020; Zhuo et al., 2013). This long exposure to Iso also likely brought a general down regulation of beta-adrenergic receptors leaving non-receptor-mediated Iso effects play a significant part here.

The reason for the posterior wall being more sensitive to Iso than the anterior one is not clear. Differences in LV regional beta-receptor sub-types density have been reported before. Greater apical beta-adrenergic receptor density or responsiveness has been described in humans, dogs, rats, cats and rabbit hearts (Mori et al., 1993; Kawano, Okada & Yano, 2003; Heather et al., 2009; Lathers, Levin & Spivey, 1986; Mantravadi et al., 2007). Use of Iso infusion to create infarct-like damages and HF or for inducing Takotsubo-like syndrome selectively targets the LV apex (Shao et al., 2013b; Lathers, Levin & Spivey, 1986; Mantravadi et al., 2007; Paur et al., 2012; Rona, 1959; Shao et al., 2013c). This basal-apex gradient of beta-adrenergic receptor responsiveness has thus been well-described. Here, our observation made in the PSLAX now includes an additional axis for a possible antero-posterior gradient in beta-adrenergic receptor sensitivity to sur-activation. Unfortunately, we could only describe this observation without providing satisfactory explanations. The complex 3D architecture of the myocardium may provide clues, but more studies are needed to provide new insights about this intriguing observation.

Speckle peak displacement data at baseline show that in the radial direction, posterior segments are more mobile than anterior ones (Fig. S2). This is associated with larger radial wall velocities for these posterior LV segments. Here too, it was these posterior segments that were more negatively targeted by Iso for both peak displacement and peak velocity. In the longitudinal direction, baseline values were relatively similar to posterior and anterior segments.

As illustrated in Fig. 4, velocity vectors originating from the LV base (and the apex) show a more important longitudinal component than radial. For the mid-ventricle segments, it is the opposite and the radial component of the velocity vectors is more important. Strain and SR measurements are expressed relative to a baseline position at the initiation of systole (R-wave). They do not provide, however, a clear evaluation of the changes in the direction (more radial or more longitudinal) of these vectors after Iso treatment. These vectors also provide information about LV relaxation. This illustrates the high complexity of cardiac contraction and relaxation and the difficulty to assess regional LV dynamic response using only one dimension at a time, here radial vs. longitudinal.

Reverse strain rate has been proposed as an index of diastolic function in mice (Schnelle et al., 2018). Considering that most parameters measured by STE are systolic in nature, reverse strain rate can constitute an interesting window to the kinetics of LV relaxation, at least during its passive filling. It is interesting to notice that again, the posterior wall segments in males are the ones with the more reduced reverse SR suggesting LV stiffening caused by Iso seemed to mainly target this region.

STE strain analysis has been performed in the past in Iso-treated mice. In male mice receiving Iso for either three or seven days, global radial and longitudinal strain and strain rate were reduced in PSLAX and only strain rates were reduced in SAX. When concentrating on LV wall segments in PSLAX, the authors found no regional differences suggesting that Iso effects on the LV after three or seven days were mostly global (An et al., 2016). Their dosage of Iso used was lower (5 vs. 30 mg/kg here) and duration of treatment shorter, making it difficult to make comparison with our work. It is probable that in the present study, more chronic mechanisms took place at the cellular and molecular levels, as mentioned above. Interestingly, after only three days, An and collaborators observed in their mice, increased myocardial fibrosis and hypertrophy. They did not mention if chamber dilatation was present but reported LV wall thickening. We did not observe this in our mice after three weeks of Iso (An et al., 2016).

We thus observe a clear reduction of global strain measurements in our animals and this was similar between males and females. These parameters are highly sensitive to detect cardiac dysfunction but do not provide regional information. Since most sex differences we observed were present at the regional level, global longitudinal and circumferential strains were not informative.

A Takotsubo cardiomyopathy model?

As mentioned above, isoproterenol has been used to develop animal models of Takotsubo syndrome. Takotsubo syndrome is a “transient LV dysfunction (hypokinesia, akinesia, or dyskinesia) presenting as apical, midventricular, basal, or focal ballooning” (Napp & Bauersachs, 2020). The adrenergic overstimulation is believed to be an important cause of this cardiomyopathy, which is more frequent in postmenopausal women suggesting possible protective roles for estrogens and/or for the male sex (Sachdev, Merz & Mehta, 2015).

The rat is usually the preferred animal model to study SIC. It has been studied acutely after a bolus administration of Iso or after a few days of treatment. These short regimens usually allow complete recuperation of systolic function in days or weeks following Iso (Wallner et al., 2016; Shao et al., 2013a; Ma et al., 2011; Redfors et al., 2014; Willis et al., 2015). Beta2-adrenergic receptor sarcolemmal localization was proposed for being responsible for the typical apical ballooning associated with Takotsubo cardiomyopathy (Wright et al., 2018). Since mid-ventricular and basal forms of this SIC exist, other mechanisms are involved.

One study had been conducted in mice using Iso (one single dose of 400 mg/kg) to induce a Takotsubo-like syndrome (Shao et al., 2013b). It did not result in global reduction of systolic function when assessed two hours after a bolus Iso injection. Segmental fractional wall thickening was measured in SAX view in these mice. Interestingly, two segments had their radial strain severely reduced namely the posterior wall and inferior free wall segments. These segments in SAX are part of the posterior wall in PSLAX. The four other LV segments had an increased strain to compensate in this study (Shao et al., 2013b). We did not observe this pattern of compensation in our mice.

Most attempts to reproduce a Takotsubo-like syndrome in rodents have relied so far on acute administration of Iso or catecholamines. This relies on the hypothesis that an acute surge of circulating catecholamines levels or adrenergic overstimulation are important parts of SIC etiology in patients. Known triggering factors in humans such as the death of a loved one, divorce, financial loss, diagnosis of a serious disease all have an important chronic component. In addition, circulating levels of catecholamines are seldom elevated in SIC patients (Shao et al., 2013b). This can make our study relevant to mimic human SIC since mice received for a long period of time a beta-receptor agonist. One limitation is that Iso does not stimulate alpha receptors as catecholamines do, eliminating the vasospasm component of Takotsumo cardiomyopathy (Shao et al., 2013b).

Interestingly, we showed that cessation of Iso was accompanied with improvement of systolic and diastolic function parameters. Diastolic parameters quickly returned to normal 5 weeks post-Iso. It was not the case for the systolic function. In females, the apex region strain got worse with time after initial normalization 5 weeks post-Iso. This suggests that myocardial abnormalities remain long after Iso cessation and that may lead to an eventual degradation of heart function. The reversible nature of Takotsubo syndrome in patients has been questioned for some time in the literature (Pelliccia et al., 2017). If regional LV contraction and global ejection fraction recover completely, other parameters of cardiac function can remain abnormal including LV global longitudinal strain and LV diastolic function (Dias et al., 2019).

Study limitations

Among the limitations of this study, tissue data cannot be obtained since the animals were not euthanized at the end of Iso treatment as their longitudinal follow-up was continued for several months. Any tissue data in this context would have been from LVs having a long period of recuperation after Iso treatment. It will be interesting to evaluate how sex hormones and age can influence LV response to chronic Iso in the future. In addition, the study in short axis (SAX) at the level of the apex, the midsection and the base the strain and related parameters could be informative to correlate changes observed here in PSLAX.

Conclusion

Segmental strain analysis in mice can provide information about the regional influence of a toxic cardiac insult such as Iso continuous infusion. We observed both similarities and sex-related differences in our male and female mice. Systolic and diastolic functions were negatively modulated by Iso. Differences between sexes were relatively subtle when studied by conventional echo. LV dilation was more important in males and remained present post-Iso. By STE, we observed Iso had a more global effect on the female LV whereas in males, the posterior wall was more specifically targeted. Normalization after Iso treatment was not complete for either male or female mice.

Supplemental Information

Table S1 Left ventricle morphology and function in male and female mice at baseline

BW: body weight, LV: left ventricle, EDD: end-diastolic LV diameter, ESD: end-systolic diameter, IVS: inter-ventricular septum, PW: posterior wall, RWT: relative wall thickness, FS: fractional shortening, EDV: end-diastolic volume, ESV: end-systolic volume, SV: stroke volume, EF: ejection fraction, HR: heart rate, CO: cardiac output, IVRT: isovolumetric relaxation time. Values are expressed as the mean +/- SEM. Control and Iso group comparisons were made using Student’s T-test.

Click here for additional data file.

Figure S1 Before after effects of Iso on the indexed strain time to peak (iT2P)

Radial (A–B) and longitudinal (C–D) iT2P were obtained using the parasternal long axis view. The six segments are identified. Characters colors correspond to those used in the graphs for each of these LV segments. Radial (E) and longitudinal (F) iT2P standard deviation between the six segments were calculated for each individual within each group (blue) and females (orange). Results are represented as violin plots (n = 8–10). Significance between groups was calculated with paired Student’s T-test. *: p < 0.05, **: p < 0.01 and ****: p < 0.0001 between corresponding pre-Iso and Iso animals.

Click here for additional data file.

Video S1 PSLAX loops of two male mice before and after Iso

Cine-loops of 3 cardiac cycles in PSLAX view of two male mice before osmotic micro-pump implantation (top) and after 21 days (bottom). As evidenced by these loops, LV internal area in both systole and diastole was larger after Iso treatment. General LV wall amplitude of movement was reduced.

Click here for additional data file.

Video S2 PSLAX loops of two female mice before and after Iso

Cine-loops of 3 cardiac cycles in PSLAX view of two female mice before osmotic micro-pump implantation (top) and after 21 days (bottom). As evidenced by these loops, LV internal area in both systole and diastole was larger after Iso treatment. General LV wall amplitude of movement was reduced. In the bottom right panel, the posterior LV was almost akinetic.

Click here for additional data file.

Figure S2 Before-after effects of Iso on speckle displacement and velocity

Ant: anterior, Post: posterior, SW: septal wall and FW: free wall. Males are represented on the left panels and females on the right. Results are represented as violin plots (n=8-10). Inner black lines represent quartiles of the data. Significance between groups was calculated with paired Student’s T-test. *: p < 0.05, **: p < 0.01 and ***: p < 0.001 between corresponding pre-Iso and Iso animals.

Click here for additional data file.

Supplemental Information 1 Author checklist

Click here for additional data file.

Supplemental Information 2 Conventional echo data

Complete echo data obtained for the study.

Click here for additional data file.

Supplemental Information 3 Speckle-tracking echo data

Data for Figures 1, 3, 4 and S1.

Click here for additional data file.

The authors want to acknowledge the technical help of Ms. Marine Clisson and Mr. Thomas Couët.

Additional Information and Declarations

Competing Interests

Author Contributions

Animal Ethics

Data Availability

The authors declare there are no competing interests.

Elisabeth Walsh-Wilkinson conceived and designed the experiments, performed the experiments, analyzed the data, prepared figures and/or tables, and approved the final draft.

Marie Arsenault conceived and designed the experiments, authored or reviewed drafts of the paper, and approved the final draft.

Jacques Couet conceived and designed the experiments, performed the experiments, analyzed the data, prepared figures and/or tables, authored or reviewed drafts of the paper, and approved the final draft.

The following information was supplied relating to ethical approvals (i.e., approving body and any reference numbers):

The protocol was approved by the Université Laval’s animal protection committee and followed the recommendations of the Canadian Council on Laboratory Animal Care (protocol #2019-360, VRR-19-075).

The following information was supplied regarding data availability:

All data for the different figures are available in the Supplemental Files.

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
