# Peer review of "Segmental analysis by speckle-tracking echocardiography of the left ventricle response to isoproterenol in male and female mice"

_PeerJ, doi:10.7717/peerj.11085_

## Round 0.1 · original submission · Major Revisions

While all three reviewers are of the view that your paper is well presented and clear, all have reservations. These include questions regarding the statistical analysis of the data (raised by reviewer 1), some issues about the methodological analysis (reviewer-2) and comparison with Grant et al 2020 (reviewer-3).

You should address each point raised by all reviewers clearly and fully; this may include the need for extra data (points 1, 2, 3, 5 and 6 of reviewer-2) and a need to remove repetition from the Results and Discussion sections.

I hope you will agree that these points will make your manuscript stronger. All reviewers acknowledge the interesting nature of the study.

Reviewer 1 ·

Basic reporting

The manuscript is clear and unambiguous, with a few grammatical and spelling errors that need to be corrected before acceptance.

The authors have cited relevant literature; however, some important references addressing sex differences in isoproterenol-induced cardiac dysfunction are missing e.g. the work of Angela Woodiwiss group. Also the reference Grant et al. 2020 should be updated to the published peer-reviewed article, instead of the preprint version.

Article is structured professionally with relevant data shared.

Experimental design

The primary research focus is within the scope of the journal.

The manuscript addresses an important gap in knowledge. The global effect of chronic isoproterenol administration on the heart has been reported previously, but reporting the segmental effects of isoproterenol is new. The authors addressed this gap in knowledge rigorously and described their methods in sufficient details.

Validity of the findings

Data presentation and interpretation:
1- The tables should show the interaction p value in addition to the Sex and Iso effects.
2- It seems that isoproterenol increased the LV mass in male mice only (from 86 to 95 mg, almost a 10% increase). Although this was not significant with this sample size and high variability, this needs to be addressed.
3- The data and the statistical analysis in Figure 1 do not address the sex differences. The data for each segment should be compared between males and females in parallel and the Two-Way ANOVA should be used here, similar to the Tables. Sex differences cannot be properly addressed without Two-Way ANOVA.
4- The authors used the Pre- to Post- comparison in the Figures, in contrast to the Tables where the used control vs. treatment. There should be an explanation.
Discussion and Conclusions:
The Discussion has a lot of repetitions from the Results section. The authors should use the Discussion to compare their findings with prior research and critically evaluate/speculate on the differences and similarities.

Additional comments

In general, the manuscript advances this particular area of research by showing segment-specific differences in the response to isoproterenol. The major limitation, as stated by the authors, is the lack of any tissue-driven data e.g. molecular markers of remodeling, histopathology,...etc. It is not hard to repeat the mouse experiments for the purpose of obtaining these tissues (these are regular C57Bl/6J mice, not very expensive). Without these tissue-level data, this reviewer thinks that the amount of data in this manuscript commensurate with a short communication, not a full length article.

Reviewer 2 ·

Basic reporting

The author gave a clear and straightforward background introduction. The structure of this manuscript is professional. The results provided by the author was almost self-contained. But some sentence in the manuscript is a little bit confusing, like "Anterior segments were also reduced in females.". Please revise them and give more detailed descriptions.

Experimental design

The author continuously infused the mice with ISO for 3 weeks and found that there were decreased systolic and diastolic cardiac function and there was a gender difference. The research design was good and the description of the methods was detailed enough.

Validity of the findings

According to the results, the author found that the strain analysis could identify regions of LV that was more negatively affected by ISO. They also mentioned that continuously infused the mice with ISO could build a model for Takosubo cardiomyopathy.
All underlying data have been provided and conclusions are well stated.

Additional comments

1. The range for HR (400-550 BPM) is relatively large. Please provide the HR for speckle tracking analysis (male and female, before and after ISO). HR might also affect the strain and strain rate.

2. Please provided the BW and/or TL corrected LV wall thickness, LV diameter/volume, CO, and SV, which would make the data more persuasive for mice with significantly different BW.

3. How about the systolic and diastolic cardiac function when using the ISO (before stop)? Did the systolic function increase or decrease after continuously infusing the ISO after 1 week, 2 weeks, or 3 weeks?

4. Have any other experiments been done to prove the decrease of systolic and diastolic cardiac function, like hemodynamic analysis?

5. When doing the speckle tracking analysis, please provide the time to peak for strain and strain rate and the standard deviation of time to peak for different segments. This parameter is critical for evaluating cardiac synchronization and represents the systolic function as well. The asynchrony could be proved in videos S1 and S2.

6. The intra- and inter-observer variability should be added for both speckle tracking and conventional echo analysis.

7. In line 106, the author mentioned that “2D echo B-mode loops were acquired from the LV PSLAX”. Should it be “PSLAX and SAX”?

8. In line 182, the author mentioned that “sometimes showed features of an infarct in the posterior wall near”. Please give more description about that, and please provide more evidence, like ECG, for infarction, if possible.

Reviewer 3 ·

Basic reporting

The manuscript is mostly well written, but suffers from numerous typological errors, including in the conclusion.

Experimental design

The echo recordings are of high quality and the animal model seems well maintained and may have clinical relevence.

The mice were not euthanised and so no morphological or biochemical data were collected.

The research described are interesting, but very similar in nature to those of Grant et al 2020. That paper used a mouse ISO infusion model for 14 days in female and males. They find functionally similar differences in both sexes.

Grant et al conclude that mice do not replicate the sexual dimorphism of humans and so may not be a relevant model for the disease. There may be a slight reduction in ejection fraction, but this was opposite to human, where males were are more severely affected.

It is unclear therefore what additional information is gained by this study.

Validity of the findings

No comment

Additional comments

The data are of high methodological quality, are analysed to a high standard data and are statistically valid. The animal model seems to replicate that previously published. The novelty of the data are the main reservation.

---

## Round 0.2 · accepted · Accept

Thank you for addressing the points raised carefully. I am now happy to recommend acceptance of your article.

Reviewer 1 ·

Basic reporting

The manuscript is clear

Experimental design

Good experimental design

Validity of the findings

Data are clearly presented

Additional comments

The authors have responded to most of the raised critiques, The additional follow-up data are interesting and added novelty to the manuscript. The dependence on imaging as the only technique used in the manuscript is the main drawback, but it was addressed by the authors as a limitation.

Reviewer 2 ·

Basic reporting

The manuscript is clear and unambiguous after revision. No further comment.

Experimental design

No further comment.

Validity of the findings

No comment.

Additional comments

For the paper “Segmental analysis by speckle-tracking echocardiography of the left ventricle response to isoproterenol in male and female mice”, the authors addressed my concerns in my review by performing more data analysis and giving more explanation and discussion. I recommend acceptance for publication.